# Yield, lodging, and water use efficiency of Tef [*Eragrostis tef* (zucc) Trotter] in response to carbonized rice husk application under variable moisture condition

Mekonnen Gebru Tekle[1,2]*, Getachew Alemayehu[2], Yayeh Bitew[2]

**1** College of Agriculture and Natural Resource Management, Wolkite University, Horticulture, Wolkite, Gurage, Ethiopia, **2** Department of Plant Sciences, College of Agriculture and Environmental Science, Bahir Dar University, Bahir Dar, Ethiopia

☯ These authors contributed equally to this work.
* mekugebru@gmail.com

**Data Availability Statement:** All relevant data are within the manuscript and its Supporting information files.

## Abstract

Terminal drought and lodging are among the major yield-limiting factors for tef cultivation in the highly weathered soils of the Ethiopian highlands. Therefore, a study was conducted to assess the yield and lodging responses of tef to varying moisture depletion levels (MDL) and the application of carbonized rice husk (CRH). A two-year 4×4 factorial experiment with 20, 35, 55, and 75% MDL and 0, 291, 582, and 873 kg ha$^{-1}$ of CRH was laid out in a split-plot design, with each treatment replicated four times. The pooled mean ANOVA showed leaf area index (LAI) and lodging index (LI) were not significantly influenced by the main and interaction effects of MDL and CRH (p > 0.05); however, individual year ANOVA showed that both LI and LAI were influenced by the interaction of MDL and CRH (p<0.05) in 2021 and 2022, respectively. The lowest LI (19.7%) was obtained from the application of 873 kg CRH ha$^{-1}$, followed by 20.6% from 582 kg CRH ha$^{-1}$ in 2022. A 20.7% LI reduction was recorded in 2022 compared to 2021. Tef plant height and number of tillers per plant were significantly affected by MDL at p<0.05 and p<0.01, respectively, but not by CRH and its interaction with MDL. The effect of MDL was significant on tef HI (p<0.01) but not on traits including grain yield, straw yield, and water use efficiency. In conclusion, the pooled mean analysis result showed that, though there was no significant difference in yield, tef irrigated at 55% MDL provided a maximum HI of 33.8%, which was 6.21% more than the control, and increased the level of lodging resistance with a LI of 31.9%, which was next to 75% MDL with 582 kg ha$^{-1}$ CRH. The authors suggested that the research should further be verified across locations for wide application.

## 1. Introduction

Drought and lodging are among the major problems in agricultural crop production in Ethiopia. Drought reduces the yield of crops, restricts growth and photosynthesis, changes

**Funding:** This research was funded by the local government, partly by the Ministry of Education and the College of Agriculture and Environmental Sciences of Bahir Dar University.

**Competing interests:** We the authors of this manuscript have declared that no competing interests exist.

morphology, decreases chlorophyll content, and affects ion balance in plants [1]. The majority of farmers in Ethiopia are engaged in smallholder, dry-land agricultural systems [2], which are vulnerable to drought caused by inter-annual rainfall variability and uptake by plants [3]. Many reports also showed the effect of drought on nutrient availability and uptake by plants [4–6]. In recent years, the increase in price in the local and international markets for food crops has increased the interest of farmers and the government in producing tef under irrigation. However, heavy fertilization and uncontrolled flooding could result in the lodging of plants and a reduction in the yield of crops. Therefore, the use of efficient water management along with an optimum rate of beneficial nutrient technologies is essential since water is becoming scarce [7] and to minimize crop yield loss [8].

Lodging, another yield limiting factor, has been reducing yield and quality of cereal crops [9,10], which is attributed to low assimilation supply during the grain filling stage [11]. Lodging is a serious problem that could generally result in a significant economic loss for the country [12]. Lodging under natural condition could cause up to 22% total Tef grain yield loss, 35% of 1000-kernel weight, and 51% of grain yield per panicle [13].

Application of growth hormones like giberelline induces lodging resistance by inhibiting plant hieght in tef [13]. The idea of improving lodging resistance through regulation of plant height was also stated by many other researchers [14–17]. Other organic compounds like paclobutrazol improve the stem strength of wheat by increasing lignin accumulation and its related enzyme activity in the basal second internode [9–18]. However, paclobutrazol caused a significant reduction in plant height [19], which could not be acceptable by farmers, as the straw has multiple benefits for their livelihood. In barley-pea intercropping, mechanical support from barley was found to be useful to reduce the lodging effect of pea [20], but it could be less applicable for tef because of severe resource competition and difficulty of harvesting.

Many other research reports stated the positive effect of silicon fertilization on the yield and lodging resistance [21,22], disease [23], and water stress tolerance [24,25] of various high silicon accumulating monocotyledon crops and sugar cane [26]. Another researcher also found significant effect of silicon in the form of potassium silicate on improving water use efficiency of maize crops [27]. However, some of the low silicon accumulating crops, such as potato [28] and tomato, were also found responsive to silicon. According to [29], the defensive property of silicon can help reduce the use of pesticides. The application of silicon increases light interception [30], P availability by increasing root exudation of organic acids that mobilize Pi in the rhizosphere and up-regulated Pi transporters [31], and avoid heavy metal toxicity [32] through co-precipitation in the soil media or regulation of the expression of metal transport gene [33], thus, biomass and grain yield of crops can be maximized. Three foliar applications of Silixol resulted in an average yield increment of about 15% in rice [34]. As per [35], silicon transcriptionally regulates sulfur and abscic acid metabolism and delays leaf senescence in barley under combined sulfur deficiency and osmotic stress. Moreover, [34] indicated the regulation of sodium transportation and distribution in maize plants growing in mild salt stress conditions. Sorghum grown under water stress conditions can provide the highest yield with a synergetic effect of irrigation and increased silicon uptake [36]. The application of silicon in the form of orthosilicic acid ($H_4SiO_4$) positively affected the growth, yield, and nutrient uptake of rice in tropical zones of Vietnam [37].

The uptake of Si is as high as that of other macronutrients like P, N and Ca in high accumulating crops like rice and sugar cane. Plants absorb Si in the form of orthosilicic acid ($H_4SiO_4$), which is not very mobile in plants. Three mechanisms of absorption are known for Si uptake: passive, active and exclusion. The first two mechanisms are common in high Si accumulating plant species. Tissue analysis from a wide variety of plants showed that silicon concentrations range from 1 to 100 g Si kg$^{-1}$ of dry weight, depending on plant species. On average monocot

plants absorb 50 to 200 kg of Si ha$^{-1}$. The largest amounts of silicon are absorbed by sugarcane (300–700 kg of Si ha$^{-1}$), rice (15–300 kg of Si ha$^{-1}$), and wheat (5–150 kg of Si ha$^{-1}$) [34].

Although Si is abundant in the earth's crust, its available form, which is orthosilicic acid (OA), is low to supply the need for plants in a highly weather soils. The upper 20 cm soil layer contains only an average of 0.1 to 1.6 kg Si ha$^{-1}$ as orthosilicic acid [38]. Inorganic silicon sources were used most commonly for research as well as production of cereals, however, they may not be eco-friendly [39] or locally available, so alternative organic sources need to be uncovered. Rice husk is considered as organic source of Si with an estimated silica content of 60 to 90% [40,41]. In addition, it is cost effective and has a capability to improve soil carbon when applied in carbonized form [42].

Tef is an annual C4 plant species that adapted to various climatic and soil conditions including arid and semi-arid regions in Ethiopia [43]. Drought tolerance mechanism of tef was found to be related to the increase in flavonoid, serine and glycine amino acids, sugars (ribose, myo-inositol), and fatty acids content [44]. The multiple use and wide adaptation make the crop preferred both by the growers and consumers in country. The pan like bread called 'injera', which is prepared from tef grain, is the staple cultural food by nearly 75% of the people in the country [45]. In recent years, tef has attracted baking factories from developed countries for making gluten-free food products [46]. The straw has also been used as a live-stock feed, cement for the construction of traditional houses. Despite multiple benefits of tef, its productivity has been hindered by extended drought [6,7] and lodging [13], among others. Conversely, the yield and lodging response of tef towards the application of silicon under variable soil moisture condition particularly in an open environment have not been verified. Therefore, the objective of this experiment was to assess the water use efficiency, yield, and lodging responses of tef to varying soil moisture content and carbonized rice husk supplement.

## 2. Materials and methods

### 2.1. Site description

A two-year field experiment was conducted to assess the productivity and lodging responses of tef to varying soil moisture content and silicon application in the two dry seasons of 2021 and 2022 at the Bahir Dar University Research Site, Kudmi, Mecha District, Northwest Ethiopia. The study site is located at 11˚23'32" to 11˚23'33" N latitude and 37˚6'42" to 37˚6'43" E longitude at an elevation of 1983 m.a.s.l. Based on the 30 years of weather data, the mean monthly minimum and maximum temperatures were 10.7 and 27.7˚C, respectively. The mean annual rainfall was 1768 mm. A summary of mean daily minimum and maximum temperatures and rainfall variability is shown graphically hereunder (Fig 1). The missing values of weather data were filled by calculating based on the means and variance of the observed data through DSSAT WeatherMan Version 4.8.0.0 [47]. Additionally, important information regarding the water quality and soil physicochemical properties is presented below (Table 1).

### 2.2. Experimental planting material

A tef variety called "*Heber-1*" (DZ-Cr-419) was used for the experiments. According to [13], Heber-1 is a very white seeded variety, which was released in 2017 by Adet Agricultural Research Centre. It is tolerant to terminal moisture stress, early set, but sensitive to lodging [13]. The variety Heber-1 is also competitive in yield with Quncho and Kora varieties that are moderately tolerant to lodging. Its altitude and annual rainfall requirements are ranging from 1500–2500 masl and 300–700 mm, respectively. The on-station yield was reported as 2.2–2.7 ton ha$^{-1}$.

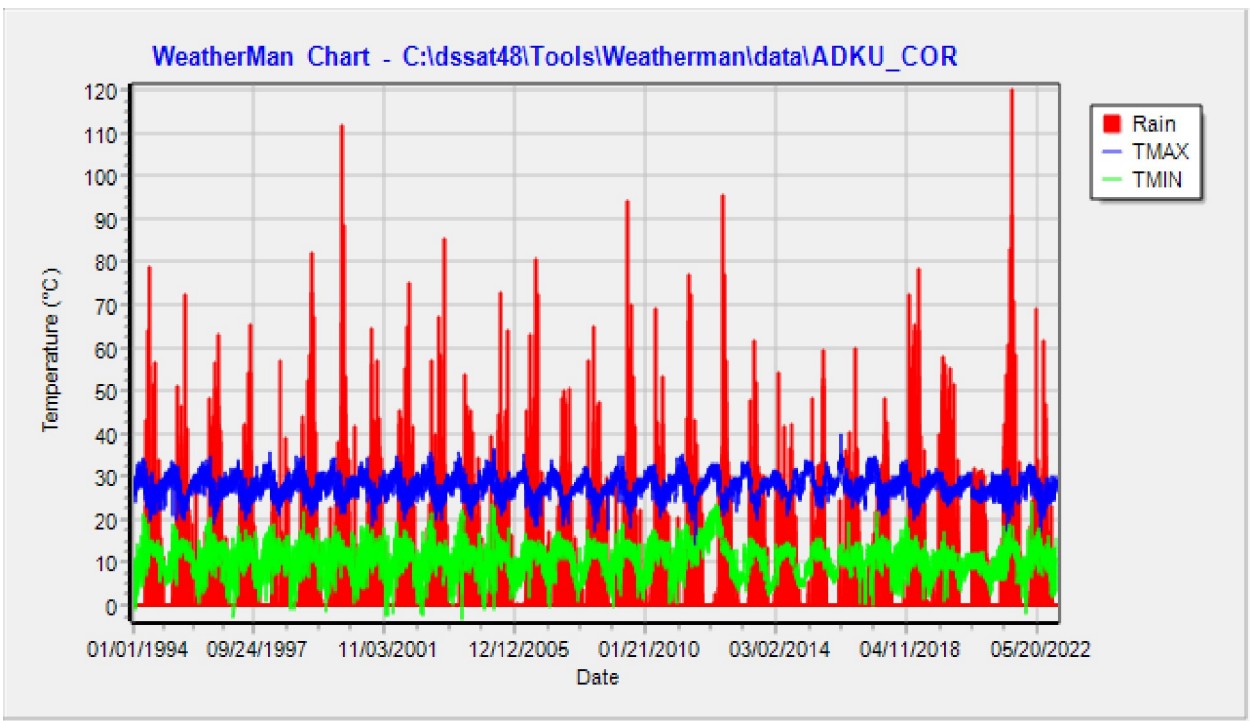

**Fig 1. Mean daily minimum and maximum temperatures and rain weather data for 01/01/1994 to 31/12/2022 (source: National Meteorological Agency of Ethiopia, Merawi Agro-meteorological station).**

Carbonized rice husk (CRH) was used as a source of Si. The material was prepared conventionally from well dried rice husk collected from nearby local market. The preparation process takes about 31 hours with the preparation procedure shown in the S1 Fig.

## 2.3. Experimental design and layout

A 4*4 factorial experiment with four soil moisture depletion levels [20% (no water stress), 35%, 55%, and 75%] and four levels of silicon in the form of carbonized rice husk (0, 291, 582, and 873 kg ha$^{-1}$) laid out in a split-plot design with four replications. The treatments for the main and sub-plot factors are presented below in Table 2. The treatments of the sub-plot factor were laid out on a net plot area of 3 m by 4 m (12 m$^2$). The main-plot factor treatments were laid out on a net experimental plot area of 13.5 m by 4 m (54 m$^2$). The space between adjacent plots and blocks will be 0.5 m and 1 m, respectively. The gross experimental plot area was 24.5 m by 62.5 m (1531.25 m$^2$), including a 2 m wide buffer zone. The field layout was provided in S2 Fig.

## 2.4. Agronomic practices

To make the soil loose, pre-planting irrigation was applied. Initially, the whole experimental plot was tilled using an ox-driven method. During the second year, each plot was tilled separately using hand-held tools, and clods were broken and levelling was done by hand. Seeds of the Heber-1 variety were sown on the experimental plots at a rate of 15 kg ha$^{-1}$ on a 12 m$^2$ plot area, which received 18 g of tef seed. The practice of sowing was carried out on well-prepared land using the row planting method at a spacing of 0.2 m.

**Table 1. Pre-planting characteristics of water, soil, and post-planting crop management data.**

| Irrigation and water quality measurement | | | | | | | |
|---|---|---|---|---|---|---|---|
| Parameter | Unit | Soil moisture depletion level | | | | Critical level | Growth stage |
| | | 20 | 35 | 55 | 75 | | |
| | | % | | | | | |
| I | days | 1–2 | 1–2 | 1–2 | 1–2 | - | Sowing to 90% emergence |
| | | 3–7 | 3–12 | 3–12 | 3–14 | - | Emergence to maturity stage |
| | | 4.8/5 | 4.6/6 | 4.7/7 | 5.2/8 | - | Growing season |
| IF | | 17 | 14 | 12 | 11 | - | All growing stage |
| Ig | mm season$^{-1}$ | 1314 | 1133 | 1161 | 1091 | - | All growing stage |
| Water Quality range over the two production seasons | | | | | | | |
| ECw | µS | 88.5–146.0 | | | | Low (not saline) | All growing stage |
| TDS | ppm | 44.9–70.8 | | | | Low | All growing stage |
| Soil physicochemical properties by soil profile layer | | | | | | | |
| | | 0–20 | 20–40 | 40–60 | | | |
| | | cm | | | | | |
| ECe | µS | 52.5 | 31.9 | 32.4 | | Low (non- salinity) [48] | |
| Soil pH | - | 5.0 | 5.4 | 5.5 | | Strongly acidic [48] | |
| BD | g/cm$^3$ | 1.23 | 1.13 | 1.10 | | | |
| OC | % | 1.96 | 1.5 | 1.38 | | | |
| Crop management by crop season (year) | | | | | | | |
| | | PD | | HD | LGP | | |
| Year | 1 | 22/01/2021 | | 06/05/2021 | 104 | | |
| | 2 | 04/01/2022 | | 13/04/2022 | 99 | | |

I, irrigation interval; IF, Irrigation frequency; Ig gross irrigation; ECw, Electrical conductivity of water; ECe, Electrical conductivity of soil; PD (HD), planting date and Harvest Date; LGP, Length of growth period; *mean of year 2021/2022

The Si treatment levels (in the form of carbonized rice husk), were randomly broadcast and incorporated into each experimental (S3 Fig) plot following the standard procedure provided for split plot design [49]. The application was done just before sowing with the broadcast incorporation method and thoroughly mixed with a digging hoe. Each plot received the recommended amounts of nitrogen (40 kg ha$^{-1}$) and phosphorous (60 kg ha$^{-1}$) in the form of urea and triple superphosphate (TSP), respectively. For good plant growth, urea was applied in a split application with 1/3 at planting and 2/3 at stem elongation, whereas TSP was applied as a band application at sowing. Other agronomic practices, including weeding and pest control, were done as per the recommendation provided for tef production in the study area [50]. Irrigation treatment was commenced when the seedlings fully emerged (8 DAP). The soil moisture content was monitored with the help of a soil moisture meter (Delta-T device,

**Table 2. Main- and sub-plot treatments used in a split-plot design.**

| S.N | Main-plot factor levels [MDL (%)] | Sub-plot factor levels [CRH (kg ha$^{-1}$)] |
|---|---|---|
| 1 | 20 (no water stress) | 0 |
| 2 | 35 | 291 |
| 3 | 55 | 582 |
| 4 | 75 | 873 |

MDL, water depletion level; CRH, Carbonized rice husk

Cambridge, UK). The soil was refilled to field capacity from the point of depletion of the corresponding treatments. The total seasonal irrigation depth, frequency, and interval of application were presented in previous table (Table 1).

## 2.5. Data collection

Plant height (PH) and panicle length (PL) were measured from the same ten plant samples, which were selected randomly from the net plot area. Plant height was measured from the base of the stem to the tip of the panicle, while the PL was measured from the node, where the first panicle branch emerges to the panicle tip by a hand-held measuring tape. Leaf area index (LAI) was taken at mid-season stage with the help of a canopy analyser (LI-COR Biosciences, LAI-2250, PCH-4750, made in the USA). The number of tillers per plot was counted at maturity stage from ten randomly selected plants in the net plot area. Plant growth during the period of water deficit was determined by calculating the relative growth rate (RGR) on the basis of PH measured every fortnight (Eq 1).

$$RGR = \frac{PH_1 - PH_o}{PH_o} \tag{1}$$

where $PH_1$ and $PH_o$ represent the PH (cm) measured at consecutive periods, with $PH_o$ the initial measurement and $PH_1$ as the next measurement.

The physiological parameters of water use efficiency (WUE) were calculated from grain yield adjusted for moisture content and the amount of water applied through irrigation (Eq 2). Chlorophyll content was measured every fortnight using a chlorophyll meter (SPAD-502 Plus, KONICA MINOLTA, INC, made in Japan) non-destructively on the leaves of five sample plants.

$$WUE = GY_{adj}/I_g \tag{2}$$

where WUE is water use efficiency; GYadj is adjusted grain yield (kg ha$^{-1}$); and $I_g$ is gross applied irrigation (m$^3$).

Above-ground biomass of tef was harvested by throwing a 1 m$^2$ quadrant to the net plot area and sun dried until constant weight reached. The grains were separated manually from the straw and husk. The grains yield (GY) and above ground biomass yield (ABY) were measured using portable digital balance with 0.01 measurement accuracy and converted to hectare basis. Adjustment in GY to 12.5% moisture content was done using Eq 3.

The ABY was measured by weighing the sun-dried above-ground harvested samples and converting them to kg ha$^{-1}$. The harvest index (HI) was calculated as the ratio of GY to ABY and expressed as a percentage, whereas the straw yield was considered the difference between ABY and GY. The lodging index (LI) was recorded by monitoring the degree of plant stand inclination towards the ground on the whole plot level with a scale of zero to five, where 0 represented 0% and 5 represented 100% plant lodging [51].

$$GY_{adj} = GY_m * \left( (100 - MC) \Big/ (100 - 12.5) \right) \tag{3}$$

Where, GYadj is adjusted grain yield (kg ha$^{-1}$) at 12.5% moisture content; $GY_m$ is the actual grain yield; MC is the actual moisture content of measured grain yield (%).

## 2.6. Data analysis

R statistical software version 4.1.1 was used to analyse the data [52]. The Shapiro-Wilk testing approach was used to determine the normality of the data distribution before performing the

analysis of variance (ANOVA) [53]. The ANOVA was performed using a statistical package called 'doebioresearch', which was built in version 4.1.3 for split-plot design analysis. When there was a significant difference, the general linear model's process was followed, and means separation was carried out using the least significant difference procedure at the 0.05 probability level. Models from one-, two-, and three-way ANOVA were fitted to the corresponding experimental design. Soil moisture depletion level and rate of CHR were treated as fixed variables in the data analysis, while year and replication were treated as random effects.

## 3. Result and discussion

### 3.1. Lodging and leaf area indices

The effect of moisture depletion level and silicon rate on the lodging response of tef is presented below (Table 3). The combined analysis result over the two years showed that lodging and leaf area index of tef were not significantly influenced by either the main or interaction of

**Table 3. The main and interaction effect of soil moisture depletion level and carbonized rice husk rate on lodging and leaf area index of tef.**

| Treatment | | LI | | | LAI | | |
|---|---|---|---|---|---|---|---|
| | | 2021 | 2022 | COY | 2021 | 2022 | COY |
| CRH (0) | Control (20) | 48.8[a-d] | 41.3[a] | 45.0[a] | 3.93[ab] | 2.83[a] | 3.38[a] |
| | 35 | 45.0[a-d] | 30.0[bc] | 37.5[abc] | 3.30[ab] | 1.96[fgh] | 2.6368[abc] |
| | 55 | 45.0[a-d] | 30.0[bc] | 31.9[bc] | 3.7[ab] | 2.15[d-g] | 2.91[abc] |
| | 75 | 45.0[a-d] | 33.75[ab] | 39.4[ab] | 3.2[ab] | 2.511[a-d] | 2.84[abc] |
| CRH (291) | Control (20) | 45.0[a-d] | 30.0[bc] | 37.5[abc] | 3.26[ab] | 2.64[ab] | 2.95[abc] |
| | 35 | 37.5[cd] | 30.0[bc] | 33.8[bc] | 3.0[b] | 1.77[h] | 2.40[bc] |
| | 55 | 33.8[d] | 26.3[bcd] | 35.6[abc] | 3.1[b] | 2.21[def] | 2.64[abc] |
| | 75 | 56.3[ab] | 26.3[bcd] | 41.3[ab] | 2.4[b] | 2.13[e-h] | 2.28[c] |
| CRH (582) | Control (20) | 45.0[a-d] | 26.3[bcd] | 35.6[abc] | 3.8[ab] | 2.512[a-d] | 3.16[ab] |
| | 35 | 60.0[a] | 22.5[cde] | 41.3[ab] | 3.92[ab] | 2.23[def] | 3.08[abc] |
| | 55 | 52.5[abc] | 15.00[e] | 33.8[bc] | 3.5[ab] | 1.80[gh] | 2.637[abc] |
| | 75 | 37.5[cd] | 18.75[de] | 28.1[c] | 4.0[ab] | 2.37[b-e] | 3.19[ab] |
| CRH (873) | Control (20) | 41.3[bcd] | 26.3[bcd] | 33.8[bc] | 2.7[b] | 2.27[c-f] | 2.48[bc] |
| | 35 | 52.5[abc] | 15.00[e] | 33.8[bc] | 4.1[ab] | 2.03[e-h] | 3.04[abc] |
| | 55 | 45.0[a-d] | 18.75[de] | 31.9[bc] | 4.8[a] | 2.09[e-h] | 3.44[a] |
| | 75 | 48.75[a-d] | 18.75[de] | 33.8[bc] | 2.9[b] | 2.60[abc] | 2.73[abc] |
| LSD.05 | | 17.12 | 10.18 | 9.791 | 1.631 | 0.367 | 0.822 |
| Significance (ANOVA) | | | | | | | |
| Y | | - | - | ** | - | - | ** |
| MDL | | ns | ns | ns | ns | ns | ns |
| CRH | | ns | ns | ns | ns | ns | ns |
| Y*CRH | | - | - | *** | - | - | ns |
| Y*MDL | | - | - | ns | - | - | ns |
| MDL*CRH | | * | ns | ns | ns | * | ns |
| [1]CV (%) | | 25.9 | 35.4 | 29.0 | 42.2 | 41.6 | 43.0 |
| [2]CV (%) | | 25.1 | 27.8 | 27.4 | 32.8 | 11.3 | 28.8 |

Y, Year; MDL, moisture depletion level; CRH, carbonized rice husk in kg ha[-1]; Y*CRH, interaction of year and carbonized rice husk; Y*MDL; year*moisture depletion level; MDL*CRH, interaction of moisture depletion and carbonized rice husk; ANOVA, analysis of variance; CV, coefficient of variation for the main (MDL) and sub-plot (CRH) factors; ns, non-significant; ***, means are significantly different at p<0.001; **, means are significantly different at p<0.01; *, means are significantly different at p<0.05

water depletion level and CRH application rate (p>0.05). However, based on the individual-year analysis, LI and LAI were significantly (p<0.05) influenced by the interaction of MDL and CRH in 2021 and 2022, respectively. The effect of CRH on LI was significantly (p<0.001) different over two study years, but not significant on the LAI (p>0.05). The overall mean values of both indices were significantly higher (p<0.01) in 2021 than in the 2022 season. The lowest LI of 19.7%, followed by 20.6%, was recorded in 2022 from the application of 873 kg ha⁻¹ CHR, and 582 kg ha⁻¹, respectively. Based on the overall mean recorded values over the two years, it showed that the LI of tef had decreased by 20.7% in 2022 as compared to 2021.

The beneficial effect of Si application on lodging of monocots, including tef, was in line with different research results [54–59]. However, our result had a little deviation from [59] in terms of rate, which had a significant effect on tef lodging and thus might be due to the difference in Si source and rate of release that determine the availability and uptake of Si content by plants. The significant variation in LI due to the effect of CRH application over the years could probably be due to the residual effect of CRH, which was supplied in the previous cropping season and causes relatively shorter PH and pre- and post-harvest management practices. The non-significant effect of treatments based on the COY might be due to the slow release of available silicon from CHR.

## 3.2. Plant height

The response of tef plant height to different soil moisture depletion levels, silicon rates, and their interactions is provided in Table 4. Plant height was significantly (P<0.05) affected by the main effect of MDL. However, it was not influenced by the main effect of CRH and its interaction with MDL (P>0.05). The maximum PH (102.8 cm) was measured from the 20% MDL treatment, but it was not significantly different from the treatment that received irrigation at 35% MDL, which provided a PH of 101.0 cm. There was a 5.7% difference in PH between the 20 and 75% MDL. The smallest difference (0.1%) was found between 55% and 75% MDL. A separate analysis of the yearly plant height result showed the presence of variation across production seasons. The mean plant height (112.4 cm), which was measured in 2021, was significantly (p<0.01) higher than the PH (86.8 cm) of 2022. The difference in PH among water depletion levels might be due to moisture stress that occurred during the early stages of crop development. This was in agreement with the report that showed a stunted growth of field crop plants under critical soil moisture conditions [60].

## 3.3. Relative growth rate

The effect of water depletion level on the relative growth rate of the tef crop was presented hereunder (Fig 2). The growth rate curve for all four treatments followed a similar trend. Initially, the RGR was slow, then increased sharply between 43 and 71 DAP, and then decreased with a similar pattern. Despite the slow start of RGR from the control treatment (20% MDL), it showed the fastest increment during the second period as compared to the other treatments. The fourth treatment (75% MDL) had shown relatively the fastest RGR of any other treatment, but it slowed during the second phase of growth. The maximum RGR (1.66) was observed from 20% MDL, while the lowest one (1.36) was observed from 75% MDL.

## 3.4. Panicle length

Panicle length is considered among the most important lodging and yield determinants. The result indicating the main and interaction effects of MDL and CRH over individual and combined analyses over years is presented in Table 5. This trait responded significantly (p<0.1) to different water depletion levels, but the effect of different CRH application rates did not

**Table 4. The effect of soil moisture depletion and the application of silicon on tef plant height (cm).**

| Treatment | | CRH (kg ha$^{-1}$) | | | | | LSD$_{.05}$ | CV (%) |
|---|---|---|---|---|---|---|---|---|
| | | 0 | 291 | 582 | 873 | Mean | | |
| MDL (%) | Control (20) | 102.3[a] | 102.3[a] | 102.2[a] | 104.3[a] | **102.8[a]** | **4.954** | |
| | 35 | 99.5[a] | 99.4[a] | 101.7[a] | 103.4[a] | **101.0[ab]** | **4.954** | |
| | 55 | 97.3[a] | 100.3[a] | 97.6[a] | 94.5[a] | **97.4[b]** | **4.954** | |
| | 75 | 97.4[a] | 94.8[a] | 98.1[a] | 98.8[a] | **97.3[b]** | **4.954** | |
| | **Mean** | **99.1** | **99.2** | **99.9** | **100.2** | **99.6** | **ns** | **5.0** |
| | **LSD0$_{.05}$** | | | | | **11.8** | | |
| 2021 | Control (20) | 113.6[ab] | 110.7[ab] | 111.4[ab] | 113.2[ab] | **112.2[a]** | **5.71** | |
| | 35 | 111.1[ab] | 113.1[ab] | 114.6[ab] | 114.0[ab] | **113.2[a]** | **5.71** | |
| | 55 | 113.4[ab] | 114.8[a] | 111.9[ab] | 109.1[b] | **112.3[a]** | **5.71** | |
| | 75 | 109.8[ab] | 111.5[ab] | 113.2[ab] | 113.5[ab] | **112.0[a]** | **5.71** | |
| | **Mean** | **112.0** | **112.5** | **112.8** | **112.4** | **112.4[a]** | **2.86** | **3.5** |
| | **LSD0$_{.05}$** | | | | | **4.30** | | |
| | **CV (%)** | | | | | **4.78** | | |
| 2022 | Control (20) | 91.1[a-d] | 94.0[ab] | 93.0[abc] | 95.4[a] | **93.4[a]** | **8.3** | |
| | 35 | 87.9[a-f] | 85.8[b-g] | 88.9[a-e] | 92.8[abc] | **88.8[ab]** | **8.3** | |
| | 55 | 81.1[efg] | 85.8[b-g] | 83.3[d-g] | 79.9[fg] | **82.5[b]** | **8.3** | |
| | 75 | 85.0[c-g] | 78.1[g] | 83.1[d-g] | 84.1[d-g] | **82.6[b]** | **8.3** | |
| | **Mean** | **86.3** | **85.9** | **87.1** | **88.0** | **86.8[b]** | **ns** | **6.7** |
| | **LSD0$_{.05}$** | | | | | **8.47** | | |
| | **CV (%)** | | | | | **12.2** | | |
| **Significance (ANOVA)** | | | | | | | | |
| **Year** | | ** | ** | ** | ** | ** | | |
| **Y*MDL** | | - | - | - | - | . | | |
| **Y*CRH** | | ns | ns | ns | ns | ns | | |
| **MDL*CRH** | | ns | ns | ns | ns | ns | | |
| **Y*MDL*CRH** | | ns | ns | ns | ns | ns | | |
| **CV (%)** | | | | | | **8.4** | | |

MDL, moisture depletion level; COY, combined analysis over years; LSD$_{0.05}$, least significance difference at 0.05 level of significance; CV, coefficient of variation; Y, year; Y*MDL, interaction of year and moisture depletion level; Y*CRH, interaction of year and silicon rate; MDL*CRH, interaction of MDL and CRH rate; Y*MDL*CRH, interaction of Y, MDL and CRH; ns, non-significance difference

influence PL significantly (p > 0.05). The interaction of both the main and sub-plot factors also did not significantly (P>0.05) affect PL. The overall mean PL measured in 2021 was significantly higher than the value measured in 2022 (p<0.01). The mean PL collected in 2021 was 30.3% higher than the mean PL in 2022. This might be due to the difference in pre- and post-harvest crop management and weather conditions [61].

### 3.5. Number of tillers per plant

The number of tillers per plant was significantly influenced by the main effect of soil moisture depletion level (p<0.01) (Table 6). However, the effect of CRH and the interaction with MDL were not significant on the NTPP of tef (p > 0.05). The maximum NTPP (1.71) was recorded from 55% MDL, but the value was indifferent as compared to 20% MDL, which provided a mean NTTP of 1.70. The fewest NTPP (1.20) was recorded from 35% MDL, but was not statistically different from 75% MDL (1.30). The variation in number of tillers per plant was in

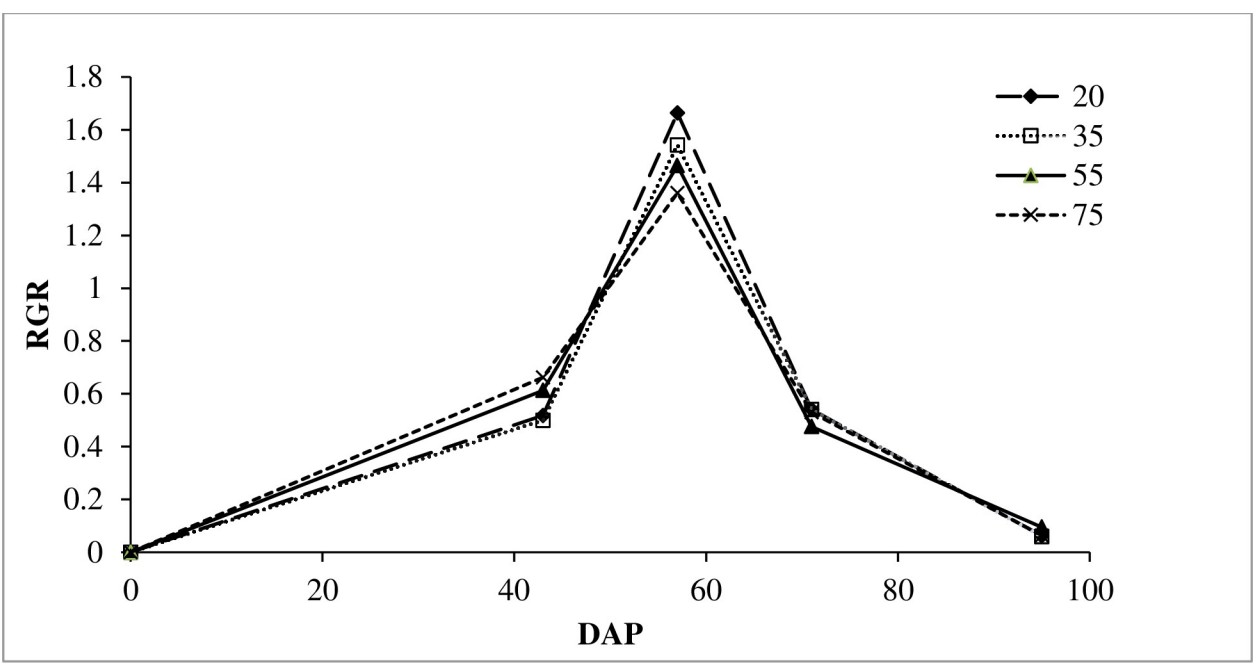

**Fig 2. The effect of moisture depletion levels on the relative growth rate of tef.** The 20, 35, 55, and 75 are percentage of moisture depletion levels. DAP is Days After Planting; RGR is Relative Growth Rate.

agreement with the result [62], who stated the difference in response of genotypes under optimal and sub-optimal moisture conditions in terms of the number of productive tillers.

## 3.6. Grain and straw yields

The combined analysis over the two growing periods (Table 7) showed that soil moisture depletion level, CRH application rate, and their interaction effect on both GY and SY were not significant (p<0.05). However, the mean GY and SY were significantly higher (p<0.001) in 2021 than in 2022. In 2021, the mean yield from the main-plot treatments (MDL) ranged from 2279–2460 kg ha$^{-1}$, whereas in 2022, it was between 1169.8 and 1227.9 kg ha$^{-1}$. The overall mean GY measured in 2021 was higher than the GY in 2022 by about 99.7%. As compared to the on-farm mean national yield, the GY measured in 2021 showed a 39.4% increment, but in 2022, the GY decreased by 29.5%. Based on the result of COY, a similar trend was observed for tef SY. The mean SY in 2021 was 121.8% higher than the SY recorded in 2022.

The difference in GY and SY across the production season could be due to the variation in inter-annual crop management. The change in the rotating crop under the low input use cultivation system during the rainy season could have contributed to variation in GY and SY across seasons. The experimental land was covered by white lupine just prior to the 2021 tef experiment, whereas Niger seed was sown as cover crop before the second season experiment. For both white lupine and Niger seed, there was no external application of any fertilizer. This might contribute to the difference in nutrient availability and uptake level on the following crop, which is tef. Researcher [63] stated that intra-season management including fertilization and crop rotation may also have had an influence on the significant difference over the two cropping years. In addition, irrigation was more frequent during 2021 and slightly less frequent in 2022 during the early stage of crop development. Another important influencing

**Table 5. The effect of moisture depletion level and silicon rate on the panicle length (cm) of tef.**

| Treatments | | CRH kg ha$^{-1}$ | | | | | LSD$_{.05}$ | CV (%) |
|---|---|---|---|---|---|---|---|---|
| | | 0 | 291 | 582 | 873 | Mean | | |
| MDL (%) | Control (20) | 37.8[a] | 38.7[a] | 38.2[a] | 38.8[a] | 38.4[a] | | |
| | 35 | 37.15[a] | 37.2[a] | 38.8[a] | 39.0[a] | 38.0[ab] | | |
| | 55 | 36.9[a] | 37.3[a] | 36.8[a] | 35.3[a] | 36.6[b] | | |
| | 75 | 36.7[a] | 35.3[a] | 36.8[a] | 37.1[a] | 36.5[b] | | |
| | Mean | 37.2 | 37.1 | 37.6 | 37.5 | 37.4 | 1.58 | 6.0 |
| | LSD$_{.05}$ | | | | | 1.69 | | |
| | CV (%) | | | | | 8.6 | | |
| 2021 | Control (20) | 41.9[ab] | 42.35[ab] | 42.2[ab] | 42.65[ab] | 42.3[a] | | |
| | 35 | 41.35[ab] | 43.20[ab] | 44.15[a] | 42.73[ab] | 42.9[a] | | |
| | 55 | 43.33[ab] | 43.23[ab] | 42.12[ab] | 41.25[b] | 42.5[a] | | |
| | 75 | 41.35[ab] | 41.53[ab] | 42.87[ab] | 41.95[ab] | 41.9[a] | | |
| | Mean | 42.98 | 42.58 | 42.84 | 42.1 | 42.4[a] | 1.43 | 4.7 |
| | LSD$_{.05}$ | | | | | 2.11 | | |
| | CV (%) | | | | | 6.216 | | |
| 2022 | Control (20) | 33.75[a-e] | 35.08[ab] | 34.20[a-d] | 34.85[abc] | 34.47[a] | | |
| | 35 | 32.95[a-e] | 31.20[def] | 33.35[a-e] | 35.33[a] | 33.21[ab] | | |
| | 55 | 30.48[ef] | 31.40[c-f] | 31.55[b-f] | 29.33[f] | 30.69[b] | | |
| | 75 | 32.13[a-f] | 29.08[f] | 30.70[def] | 32.15[a-f] | 31.01[b] | | |
| | Mean | 32.3 | 31.69 | 32.45 | 32.91 | 32.3[b] | 1.78 | 12.4 |
| | LSD$_{.05}$ | | | | | 2.97 | | |
| | CV (%) | | | | | 17.4 | | |
| **Significance of ANOVA result** | | | | | | | | |
| Y*MDL | | ns | ns | ns | ns | ns | | |
| Y*CRH | | ns | ns | ns | ns | ns | | |
| MDL*CRH | | ns | ns | ns | ns | ns | | |
| Y*MDL*CRH | | ns | ns | ns | ns | ns | | |

CV, coefficient of variation; LSD$_{.05}$; least significant difference at p = 0.05; ns, non-significant difference; Y*MDL, interaction of year and moisture depletion level; Y*CRH, interaction of year and CRH; MDL*CRH, interaction of MDL and CRH; Y*MDL*CRH, interaction of year, MDL, and CRH; means within group connected with same letter/s are not significantly different at p = 0.05 level of probability

**Table 6. The effect of soil moisture depletion level and rate of carbonized rice husk on tiller number of tillers per plant of tef crop.**

| MDL (%) | CRH (kg ha$^{-1}$) | | | | | LSD$_{.05}$ | CV (%) |
|---|---|---|---|---|---|---|---|
| | 0 | 291 | 582 | 873 | Mean | | |
| Control (20) | 1.65[ab] | 2.1[a] | 1.35[b] | 1.68[ab] | 1.70[a] | | |
| 35 | 1.15[b] | 1.15[b] | 1.28[b] | 1.18[b] | 1.20[b] | | |
| 55 | 2.2[a] | 1.55[ab] | 1.55[ab] | 1.58[ab] | 1.71[a] | | |
| 75 | 1.23[b] | 1.15[b] | 1.65[ab] | 1.23[b] | 1.30[b] | | |
| Mean | 1.55[a] | 1.46[a] | 1.46[a] | 1.41[a] | 1.50 | 0.375 | 35.3 |
| LSD$_{.05}$ | | | | | 0.25 | | |
| CV (%) | | | | | 21.0 | | |
| **Significance (ANOVA)** | | | | | | | |
| MDL | ** | ** | ns | ns | ** | | |
| MDL*CRH | ns | ns | ns | ns | ns | | |

Note that this result was based on one year data (2022); LSD$_{.05}$, least significant difference at a probability level of 0.05; MDL, soil moisture depletion level; CRH, carbonized rice husk; MDL*CRH, interaction of MDL and CRH; means connected with same letter/s are not significantly different at the p = 0.05 level of significance

**Table 7. The effect of soil moisture depletion levels (%) and carbonized rice husk rates (kg ha$^{-1}$) applied in the form of carbonized rice husk on grain and straw yields of tef.**

| Treatments | | Grain yield (kg ha$^{-1}$) | | | Straw yield (kg ha$^{-1}$) | | |
|---|---|---|---|---|---|---|---|
| | | 2021 | 2022 | COY | 2021 | 2022 | COY |
| MDL | Control (20) | 2460.6[a] | 1177.0[a] | 1818.8[a] | 5168.7[a] | 2618.8[a] | 3893.8[a] |
| | 35 | 2318.0[a] | 1184.4[a] | 1751.2[a] | 5165.4[a] | 2381.5[a] | 3773.4[a] |
| | 55 | 2444.1[a] | 1227.9[a] | 1835.9[a] | 5388.0[a] | 2192.2[a] | 3790.1[a] |
| | 75 | 2279.2[a] | 1169.8[a] | 1724.5[a] | 5220.6[a] | 2248.1[a] | 3734.4[a] |
| CRH (0) | Control (20) | 2631.0[a] | 1185.5[abc] | 1908.2[a] | 5299.8[ab] | 2709.4[ab] | 4004.6[a] |
| | 35 | 2158.3[b] | 1237.9[abc] | 1698.1[a] | 5289.3[ab] | 2511.6[abc] | 3900.4[a] |
| | 55 | 2390.5[ab] | 1149.7[bc] | 1770.1[a] | 4940.3[ab] | 2083.3[c] | 3511.8[a] |
| | 75 | 2187.5[ab] | 1176.7[bc] | 1682.1[a] | 4554.0[b] | 2343.8[abc] | 3448.9[a] |
| CRH (291) | Control (20) | 2490.3[ab] | 1137.5[bc] | 1813.9[a] | 5119.0[ab] | 2491.1[abc] | 3805.0[a] |
| | 35 | 2260.0[ab] | 1108.2[bc] | 1684.1[a] | 4995.3[ab] | 2194.8[bc] | 3595.0[a] |
| | 55 | 2616.3[ab] | 1400.7[a] | 2008.5[a] | 5845.8[a] | 2215.5[bc] | 4030.6[a] |
| | 75 | 2225.5[ab] | 1089.8[bc] | 1657.6[a] | 5342.5[ab] | 2131.6[c] | 3737.1[a] |
| CRH (582) | Control (20) | 2419.8[ab] | 1177.2[bc] | 1798.5[a] | 5150.3[ab] | 2490.7[abc] | 3820.5[a] |
| | 35 | 2435.0[ab] | 1210.1[abc] | 1822.6[a] | 5175.0[ab] | 2425.7[abc] | 3800.4[a] |
| | 55 | 2320.0[ab] | 1062.3[c] | 1691.1[a] | 5517.3[ab] | 2139.9[c] | 3828.6[a] |
| | 75 | 2427.0[ab] | 1242.1[abc] | 1834.5[a] | 5657.8[ab] | 2263.1[bc] | 3960.4[a] |
| CRH (873) | Control (20) | 2301.5[ab] | 1208.0[abc] | 1754.7[a] | 5105.8[ab] | 2784.2[a] | 3945.0[a] |
| | 35 | 2418.8[ab] | 1181.3[bc] | 1800.0[a] | 5202.0[ab] | 2393.6[abc] | 3797.8[a] |
| | 55 | 2449.5[ab] | 1298.6[ab] | 1874.1[a] | 5248.8[ab] | 2329.9[abc] | 3789.4[a] |
| | 75 | 2276.8[ab] | 1170.6[bc] | 1723.7[a] | 5328.3[ab] | 2254.0[bc] | 3791.1[a] |
| **Significance of the analysis of variance test** | | | | | | | |
| **Year** | | | | *** | | | *** |
| **MDL** | | **ns** | **ns** | **ns** | **ns** | **ns** | **ns** |
| **CRH** | | **ns** | **ns** | **ns** | **ns** | **ns** | **ns** |
| **Y*CRH** | | | | **ns** | | | **ns** |
| **Y*MDL** | | | | **ns** | | | **ns** |
| **MDL*CRH** | | **ns** | **ns** | **ns** | **ns** | **ns** | **ns** |
| **$^1$CV (%)** | | **13.9** | **26.8** | **16.5** | **15.2** | **27.4** | **19.1** |
| **$^2$CV (%)** | | **11.3** | **12.8** | **14.4** | **15.8** | **15.3** | **16.8** |

$^{1,2}$CV, coefficient of variation for the main (MDL) and sub-plot (CRH) factors;

Y, year; MDL, soil moisture depletion level; CRH, carbonized rice husk; Y*MDL, interaction of year and soil moisture depletion level; Y*CRH, year, and CRH interaction; MDL*CRH, interaction of soil moisture depletion and CRH;

*, **, ***, ns are significant at 0.05, 0.01, and 0.001 probability levels, respectively; ns is a non-significant difference;

Treatments within the group connected with the same letter are not significantly different.

factor could be the variation in the number tillage and weeding practices, which might affect the tef yield significantly.

### 3.7. Harvest index

The harvest index of tef was significantly (p<0.01) influenced by the change in the soil moisture depletion level (Table 8). The effect of MDL was significantly (p<0.001) interacted with the cropping year. However, both the main effect of CRH and its interaction with MDL on HI of tef were not significant (p>0.05). The interaction of CHR with year was also not significant

**Table 8. The combined and individual analysis result of two year experiment showing the effect soil moisture depletion levels and the application of silicon in the form of carbonized rice husk on the harvest index of tef.**

| Treatments | | CRH (kg ha$^{-1}$) | | | | Mean | LSD$_{.05}$ | CV (%) |
|---|---|---|---|---|---|---|---|---|
| | | 0 | 291 | 582 | 873 | | | |
| MDL (%) | Control (20) | 31.9[bcd] | 32.2[bcd] | 31.9[bcd] | 30.7[d] | 31.7[b] | 2.64 | |
| | 35 | 31.4[d] | 32.5[a-d] | 32.6[a-d] | 32.7[a-d] | 32.2[b] | 2.64 | |
| | 55 | 34.4[ab] | 35.0[a] | 31.8[cd] | 34.2[abc] | 33.8[a] | 2.64 | |
| | 75 | 32.9[a-d] | 31.9[bcd] | 32.9[a-d] | 32.3[bcd] | 32.5[b] | 2.64 | |
| | Mean | 32.7 | 32.9 | 32.3 | 32.5 | 32.6 | 1.32 | 8.1 |
| | LSD$_{.05}$ | | | | | 1.12 | | |
| | CV (%) | | | | | 6.6 | | |
| 2021 | Control (20) | 33.0[a] | 33.0[a] | 31.9[a] | 31.3[a] | 33.0[a] | 4.095 | |
| | 35 | 29.6[a] | 31.3[a] | 32.0[a] | 32.2[a] | 29.6[a] | 4.095 | |
| | 55 | 33.2[a] | 30.9[a] | 30.1[a] | 32.2[a] | 33.2[a] | 4.095 | |
| | 75 | 32.5[a] | 29.6[a] | 30.4[a] | 29.9[a] | 32.5[a] | 4.095 | |
| | Mean | 32.1 | 31.2 | 31.1 | 31.4 | 32.1 | 2.05 | 9.1 |
| | LSD$_{.05}$ | | | | | 2.32 | | |
| | CV (%) | | | | | 9.2 | | |
| 2022 | Control (20) | 30.8[de] | 31.4[cde] | 31.9[cde] | 30.2[e] | 31.1[d] | 3.46 | |
| | 35 | 33.2[be] | 33.7[bcd] | 33.3[b-e] | 33.2[b-e] | 33.3[c] | 3.46 | |
| | 55 | 35.6[b] | 39.1[a] | 33.4[b-e] | 36.2[ab] | 36.1[a] | 3.46 | |
| | 75 | 33.4[b-e] | 34.1[bcd] | 35.4[b] | 34.7[bc] | 34.4[b] | 3.46 | |
| | Mean | 33.2 | 34.6 | 33.5 | 33.6 | 33.7 | 1.73 | 7.2 |
| | LSD$_{.05}$ | | | | | 0.68 | | |
| | CV (%) | | | | | 2.5 | | |
| **Significance of ANOVA of the pooled mean over the two years** | | | | | | | | |
| MDL | | ns | ns | ns | ns | ** | | |
| CRH | | ns | ns | ns | ns | ns | | |
| Y*MDL | | *** | *** | ns | *** | *** | | |
| Y*CRH | | ns | ns | ns | ns | ns | | |
| MDL*CRH | | ns | ns | ns | ns | ns | | |
| Y*MDL*CRH | | | | | | | | |

CRH, carbonized rice husk; MDL, soil moisture depletion level; Y, year; LSD$_{.05}$, least significant difference at a probability of 0.05; CV, coefficient of variation; Y*MDL, Y*CRH, MDL*CRH, and Y*MDL*CRH are interactions of the defined factors; means connected by the same letter are not significantly different at a probability of 0.05

on the HI of tef. Both deficit and excessive soil moisture application levels had a significant effect on plant growth and development. Deficit irrigation causes stunted growth, which could reduce overall biomass yield, while increasing the harvest index of crops, while excessive application reduces respiration rate of plants [63]. The result was not in line with [64], who stated the sever effect of moisture stress during the late-season stage than early ones.

### 3.8. Water use efficiency of tef

The combined analysis over the year (COY) result on the effect of MDL and CRH is presented in Table 9. The result showed that the WUE of tef was not significantly influenced both by the main and interaction effects of MDL and CRH (p>0.05). The COY result indicated that the interaction of MDL and CRH was only significantly affected tef WUE at p<0.1. This also interacted with the cropping year at the same level of significance. The individual year analysis

**Table 9. The effect of soil moisture depletion and the application of silicon in the form of carbonized rice husk on tef water use efficiency (kg ha$^{-1}$ m$^{-3}$) based on the individual year and pooled mean analysis result.**

| Treatments | | CRH (kg ha$^{-1}$) | | | | Mean | LSD.$_{05}$ | $^2$CV (%) |
|---|---|---|---|---|---|---|---|---|
| | | 0 | 291 | 582 | 873 | | | |
| MDL (%) | Control (20) | 1.997$^{a-d}$ | 1.992$^{a-d}$ | 1.878$^{bcd}$ | 1.824$^{bcd}$ | 1.923$^a$ | | |
| | 35 | 1.754$^{cd}$ | 1.833$^{bcd}$ | 2.009$^{a-d}$ | 1.881$^{bcd}$ | 1.869$^a$ | | |
| | 55 | 1.949$^{a-d}$ | 2.182$^{ab}$ | 1.745$^{cd}$ | 2.112$^{abc}$ | 1.997$^a$ | | |
| | 75 | 1.944$^{a-d}$ | 2.030$^{a-d}$ | 2.294$^a$ | 1.724$^d$ | 1.998$^a$ | | |
| | Mean | 1.910$^a$ | 2.000$^a$ | 1.980$^a$ | 1.89$^a$ | 1.947 | 0.193 | 19.9 |
| | LSD.$_{05}$ | 0.386 | 0.386 | 0.386 | 0.386 | 0.393 | | |
| | $^1$CV (%) | | | | | 38.4 | | |
| 2021 | Control (20) | 3.1$^{ab}$ | 2.96$^{abc}$ | 2.73$^{a-d}$ | 2.75$^{a-d}$ | 2.9$^a$ | | |
| | 35 | 2.22$^d$ | 2.35$^{cd}$ | 2.80$^{a-d}$ | 2.58$^{a-d}$ | 2.5$^a$ | | |
| | 55 | 2.496$^{bcd}$ | 2.96$^{abc}$ | 2.20$^d$ | 2.85$^{a-d}$ | 2.6$^a$ | | |
| | 75 | 2.48$^{bcd}$ | 2.76$^{a-d}$ | 3.25$^a$ | 2.23$^d$ | 2.7$^a$ | | |
| | Mean | 2.58$^a$ | 2.757$^a$ | 2.754$^a$ | 2.60$^a$ | 2.7 | 0.347 | 18.1 |
| | LSD.$_{05}$ | | | | | 0.806 | | |
| | $^1$CV (%) | | | | | 37.7 | | |
| 2022 | Control (20) | 0.342$^c$ | 0.409$^{bc}$ | 0.408$^{bc}$ | 0.360$^c$ | 0.4$^b$ | | |
| | 35 | 0.514$^{ab}$ | 0.528$^{ab}$ | 0.470$^{abc}$ | 0.475$^{abc}$ | 0.50$^a$ | | |
| | 55 | 0.561$^a$ | 0.560$^a$ | 0.521$^{ab}$ | 0.549$^{ab}$ | 0.55$^a$ | | |
| | 75 | 0.563$^a$ | 0.522$^{ab}$ | 0.536$^{ab}$ | 0.486$^{abc}$ | 0.53$^a$ | | |
| | Mean | 0.495 | 0.504 | 0.480 | 0.470 | 0.49 | ns | 21.2 |
| | LSD.$_{05}$ | | | | | 0.10 | | |
| | $^1$CV (%) | | | | | 26.5 | | |
| **Significance of ANOVA based on the pooled mean result over the two years** | | | | | | | | |
| Year | | . | . | . | . | ns | | |
| MDL | | | | | | ns | | |
| CRH | | | | | | ns | | |
| Y*MDL | | | | | | ns | | |
| Y*CRH | | | | | | ns | | |
| MDL*CRH | | . | . | . | . | ns | | |
| Y*MDL*CRH | | . | . | . | . | ns | | |

., significant at p<0.1;

MDL, water depletion level; CRH, carbonized rice husk;

$^{1,2}$CV, coefficient of variation of main-plot (MDL) and sub-plots (CRH) factors;

LSD.05, least significant difference at the 0.05 level of probability; Y*MDL, interaction of year and water depletion level; Y*carbonized rice husk, year and silicon rate;

MDL*CRH, interaction of water depletion level and CRH; Y*MDL*CRH, interaction of year, water depletion level, and CRH

result showed that in 2021, the interaction of MDL and CRH significantly affected the WUE of the tef crop (p<0.05). In 2022, however, the WUE of tef was not affected by the interaction of MDL and CRH, but it was significantly affected by the main effect of MDL (p<0.05). In 2021, the application of 582 kg CRH ha$^{-1}$ combined with 75% of MDL showed a higher maximum WUE than any other treatment combination. In 2022, the maximum water use efficiency (0.55 kg ha$^{-1}$ m$^{-3}$) was observed on 55% MDL but was not statistically different from 75% MDL, while the lowest WUE was recorded from the control (20% MDL) treatment, with a value of 0.4 kg ha$^{-1}$ m$^{-3}$. Despite not being statistically different, the pooled mean values from the two-year data showed that 75% MDL had slightly been above all other treatments with a mean

WUE of 1.997 kg ha$^{-1}$ m$^{-3}$. The high water use efficiency value indicated a better grain yield with relatively little irrigation applied [63].

### 3.9. Chlorophyll content

The effects of water depletion level and silicon rate on the chlorophyll content of tef leaf measured at four different periods of plant growth are presented in Fig 3. The chlorophyll content of tef was significantly influenced by the level of soil water depletion at the first growth period (p<0.05). However, the effect was not significant for the remaining 3 sampling periods (p>0.05). The main effect of CRH and its interaction with MDL was not significant for all the

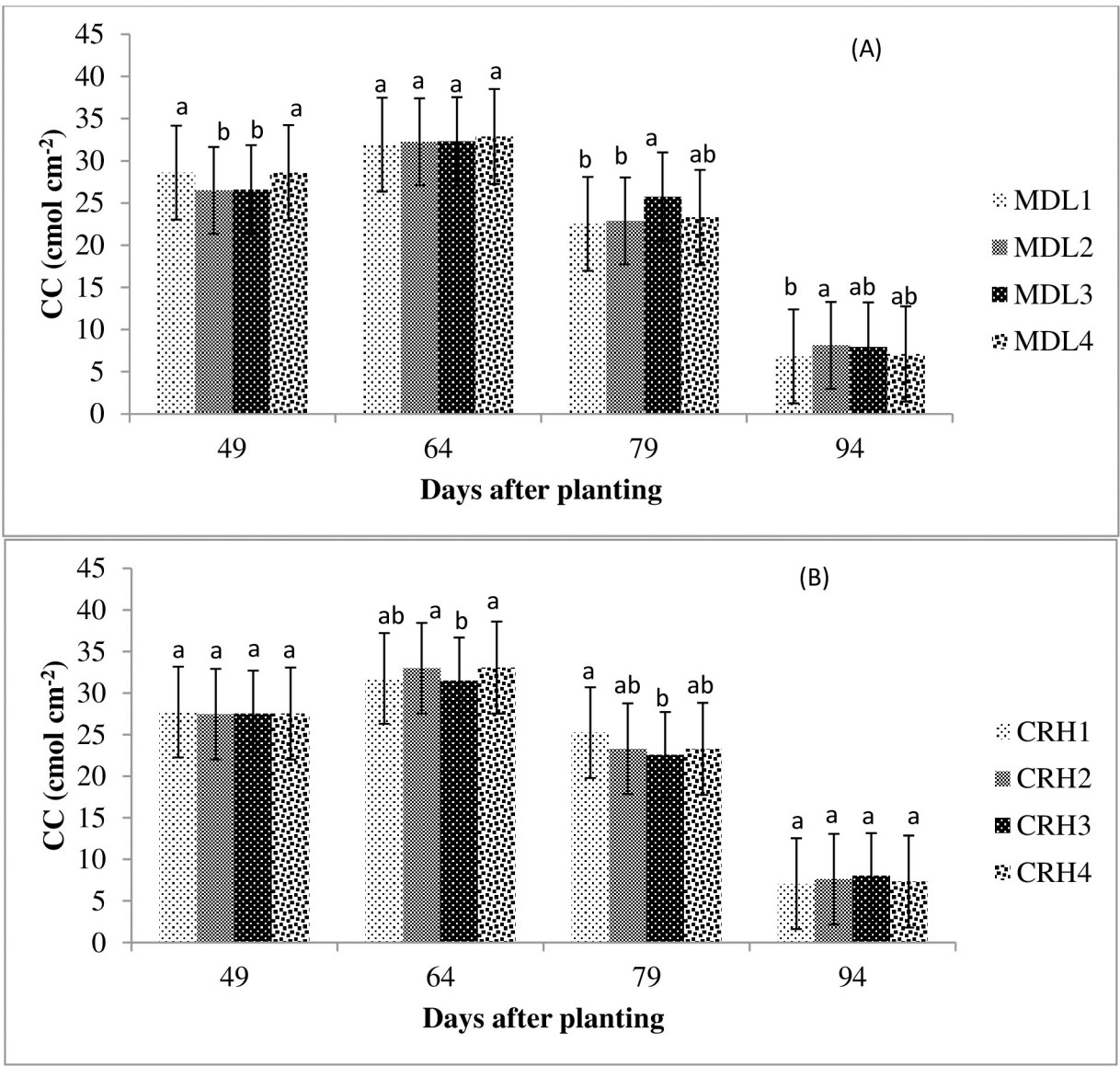

**Fig 3. Bar chart showing the effect of moisture depletion level and carbonized rice husk on chlorophyll content of tef.** (A) MDL1, MDL2, MDL3, and MDL4 with 20, 35, 55, and 75%; (B) CRH1, CRH2, CRH3, and CRH4 with 0, 291, 582, 873 kg ha$^{-1}$, respectively; the interaction of MDL and CRH was not significant (P>0.05).

periods (p>0.05). The pooled mean result also showed that CC was not influenced by both the main and interaction effects of MDL and CRH. The result was based on single year data, which was collected in 2022. It could not tell us the consistency of the effect of treatments over the season. The significant effect of MDL during the early stage of the crop might be due to the crop's sensitivity to water stress during the early growth stage [13]. The non-significant treatment differences could be attributed to the more frequent application of irrigation during the early stages of crop emergence and development rather than an extended application interval. This was in agreement with [61], who stated the presence of significant tef response to irrigation frequency at the early, vegetative and productive stages.

The bar chart showed that CC had shown an increasing trend during the first period (49–64 DAP), followed by a decreasing trend for the remaining periods. The maximum CC of 28.62 and 27.7 nmol cm$^{-2}$ was observed at 64 DAP, while the lowest CC of 6.8 and 7.1 nmol cm$^{-2}$ was observed at 94 DAP for both water depletion level and CRH application rate, respectively. This could be attributed to the change in plant physiological activity, whereby evapotranspiration and photosynthetic activities reached peaks at the end of the crop development stage and lows at the senescence stage [62].

## 4. Conclusion

Drought and lodging are among the major yield-limiting factors for tef cultivation in the highlands of Ethiopia. Tackling these problems through optimum irrigation and CRH supply could improve the yield and lodging resistance of Tef. Therefore, this study found that the moisture depletion level significantly influenced the plant height, number of tillers per plant, harvest index, and lodging resistance of tef but not the grain and straw yields. The pooled means analysis result showed that the main effect of CRH and its interaction with MDL did not significantly influenced the LI, LAI, PH, PL, GY, and SY. The application of CRH influenced the PL of tef. The effects of MDL and CRH and their interactions showed significant differences over the years. In the 2022 season, WUE was affected by the main effect of MDL, while the interaction effect of MDL and CRH was only significant at p<0.1. The maximum RGR (1.66) was observed from 20% MDL, while the lowest one (1.36) was observed from 75% MDL. Generally, it was concluded that though there was no significant difference in both grain and straw yields, tef irrigated at 55% MDL provided a maximum HI of 33.8%, which was 6.21% more than the control and increased the level of lodging resistance with a second lowest LI of 31.9% next to 75% MDL with 582 kg ha$^{-1}$ CRH. The authors suggested that the research should further be verified across locations for wide application.

## Supporting information

**S1 Fig. Preparation procedure for carbonized from a well dried rice husk.** Phase I to III shows heating process and change in the color of the husk yellow to black; phase IV is the final stage indicated by complete conversion of rice husk to black color, thus cooling takes place by spraying water to avoid conversion to the ash form.
(TIF)

**S2 Fig. Field layout of a two factor experiment in a split-plot design.**
(TIF)

**S3 Fig. Broadcast incorporation of carbonized rice husk.** a) Application by broadcasting; b) manual mixing up with the soil.
(TIF)

**S1 Data.**
(XLSX)

## Acknowledgments

The authors are grateful to research and community service office, college of agriculture and environmental sciences, Bahir dar University for their relentless efforts to the successful implementation of the SENIT project.

## Author Contributions

**Conceptualization:** Mekonnen Gebru Tekle, Getachew Alemayehu.

**Data curation:** Mekonnen Gebru Tekle.

**Formal analysis:** Mekonnen Gebru Tekle.

**Funding acquisition:** Getachew Alemayehu.

**Investigation:** Mekonnen Gebru Tekle.

**Project administration:** Mekonnen Gebru Tekle, Yayeh Bitew.

**Supervision:** Getachew Alemayehu, Yayeh Bitew.

**Validation:** Yayeh Bitew.

**Writing – original draft:** Mekonnen Gebru Tekle.

**Writing – review & editing:** Mekonnen Gebru Tekle, Yayeh Bitew.

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
