## [Decision Letter · Decision Letter 0]

10 Jul 2023

PONE-D-23-13722Yield, lodging, and water use efficiency of Tef [Eragrostis tef (zucc) Trotter] in response to carbonized rice husk application under variable moisture conditionPLOS ONE

Dear Dr. Tekle,

Thank you for submitting your manuscript to PLOS ONE. After careful consideration, we feel that it has merit but does not fully meet PLOS ONE’s publication criteria as it currently stands. Therefore, we invite you to submit a revised version of the manuscript that addresses the points raised during the review process.

We look forward to receiving your revised manuscript.

Kind regards,

Surajit Mondal, PhD

Academic Editor

PLOS ONE

5. We note that Figures S1 and S3 in your submission contain copyrighted images. All PLOS content is published under the Creative Commons Attribution License (CC BY 4.0), which means that the manuscript, images, and Supporting Information files will be freely available online, and any third party is permitted to access, download, copy, distribute, and use these materials in any way, even commercially, with proper attribution. For more information, see our copyright guidelines: http://journals.plos.org/plosone/s/licenses-and-copyright.

a. You may seek permission from the original copyright holder of Figures S1 and S3 to publish the content specifically under the CC BY 4.0 license.

Additional Editor Comments:

Both the reviewers completed their reviews and suggested 'Major Revision'. Please revise the manuscript before further reviews.

Reviewers' comments:

Reviewer's Responses to Questions

**Comments to the Author**

1. Is the manuscript technically sound, and do the data support the conclusions?

Reviewer #1: Yes

Reviewer #2: Partly

2. Has the statistical analysis been performed appropriately and rigorously? 

Reviewer #1: Yes

Reviewer #2: Yes

3. Have the authors made all data underlying the findings in their manuscript fully available?

Reviewer #1: Yes

Reviewer #2: Yes

4. Is the manuscript presented in an intelligible fashion and written in standard English?

Reviewer #1: Yes

Reviewer #2: Yes

5. Review Comments to the Author

Reviewer #1: the manuscript is an excellent work on tef, and authors had many indexes to support their opinions. the results may improve the cultivation ways of tef. However, the manuscript is like a Master thesis, not like a paper. there are some problems:

1 the Introduction part should be reorgnized to show the thoughts simply;

2 the tables are complex, it is better to simplify the tables, or change them into bar chart or line chart. In addition, the tables should be three-line tables.

3 it is better to offer the complete name of the first appeared abbreviation.

4 lines 167-168, heber-1an hiber-1, which one is right?

5 Figures in the PDF or figures sent to me are not professional ones, they should be redone.

6 the manuscript had not highly summarized experimental data and the possible relations between data.

7 there are some spoken english in the paper, it is better to correct them.

Reviewer #2: Find the attached document for my detailed comments and suggestions.

Specif comment:

The lifecycle of Tef and how it is affected by lodging (causes and the imposed treatments can be used to mitigate it) and water deficiency should be well described in the Introduction part.

6. PLOS authors have the option to publish the peer review history of their article (what does this mean?). If published, this will include your full peer review and any attached files.

Reviewer #1: No

Reviewer #2: **Yes: **Suleiman Kehinde Bello

---

## [Author Response · Author response to Decision Letter 0]

19 Oct 2023

Response to Reviewers

First of all we the authors are very grateful to all the reviewers for their valuable and detailed comments, suggestion, and questions made for the betterment of the paper. Here we tried to address the questions raised by the reviewers, while accepting all the comments.

a. Response to Academic Editors

#1. Regarding style of writing: we did our best to revise and ensure the requirement of the style of writing of the journal

#2. About grant information: this research was conducted with no funding.

#3. Repository information is included here with the revised manuscript

#4. About the map (Fig 1) which may be copyrighted: the map was created using arcGIS software for illustration purpose, but upon reducing the volume of the paper, we removed it indicate the site projection points in the text.

#5. Figures S1 and S3 are the original photos, which were taken during the research work and are original, which are owned by the corresponding author, thus we affirm that there is no copy right issue.

b. Responses to Reviewer (R#1)

Q#1. What do the terms LAI, LI, PH, and NTPP stands for?

Authors’ Response: We missed to define the abbreviations, which leads to confusion of the readers and we accepted and made a mistake and took correction.

Q#2. Does this mean that the different CRH levels were combined as a treatment? 

Authors’ Response: No. It was typing error. CRH levels were combined with moisture depletion levels as a treatment. Therefore, we made a correction on the statement.

Q#3. From which treatment does the lowest lodging index (31.9%) was recorded? 

Authors’ response: we found that the statement that we wrote lacked clarity and needs restructuring. Therefore, we made correction on the conclusion.

Q#4. How did the authors arrive at this suggestion (alternative silicon fertilizer sources) in the context of the current study?

Authors’ response: we arrived at this suggestion, based on the market analysis of for the rice husk, loading and unloading, and its preparation, the application of carbonized rice husk could be expensive to the farmers in the study site, and we believed to see other economically feasible silicon sources. However, it looks rubbish to suggest this in the context of the current study and we have changed it.

Q#5. What does the 22% yield loss comprised off? 

Authors’ Response: The 22% yield loss comprises the grain yield.

Q6. How does silicon helps in achieving increased P availability and avoid heavy metal toxity?

Silicon increases P availability by increasing root exudation of organic acids that mobilize Pi in the rhizosphare and up-regulated Pi transporters

Q7. Were the levels of carbonized rice husk based on the silicon content?

Authors’ response: No. the levels of carbonized rice husk were set based on the recommendation provided for rice crop.

Q8. Why was the water depletion level and the CRH treated as the fixed variable and replication as random variable and not the other way round? 

Authors’ response: According to Gomez and Gomez (1984), in an experimental research there are two variations, which are the within group and between group variations. The treatments are the variables that should be under the researchers control and levels are predefined at a fixed level, thus helps to detect the real variation comes from the between group variation ((fixed effect), replication is required to control the random effect that comes from the within group variation and helps to estimate the magnitude of the random error (random effect) not the treatment effect.

Q9. Regarding tables or use bar charts

For the majority of the tables, which contain both the main and interaction effects of MDL and CRH we tried to make them as simple as possible in the revised paper, additionally we use bar chart for the analysis result of chlorophyll content and its trend over growing periods.

Q10. What does COY stands for? 

Authors’ Response: COY stands for combined analysis over the years

c. Responses to Reviewer (R#2)

With due respect the reviewer provided comments and suggestion to be made, thus we accept those comments and suggestion provided to us and included to this revised manuscript.

---

## [Decision Letter · Decision Letter 1]

25 Jan 2024

Yield, lodging, and water use efficiency of Tef [Eragrostis tef (zucc) Trotter] in response to carbonized rice husk application under variable moisture condition

PONE-D-23-13722R1

Dear Dr. Tekle,

We’re pleased to inform you that your manuscript has been judged scientifically suitable for publication and will be formally accepted for publication once it meets all outstanding technical requirements.

Kind regards,

Surajit Mondal, PhD

Academic Editor

PLOS ONE

Additional Editor Comments (optional):

Reviewers' comments:

Reviewer's Responses to Questions

**Comments to the Author**

1. If the authors have adequately addressed your comments raised in a previous round of review and you feel that this manuscript is now acceptable for publication, you may indicate that here to bypass the “Comments to the Author” section, enter your conflict of interest statement in the “Confidential to Editor” section, and submit your "Accept" recommendation.

Reviewer #1: All comments have been addressed

2. Is the manuscript technically sound, and do the data support the conclusions?

Reviewer #1: (No Response)

3. Has the statistical analysis been performed appropriately and rigorously? 

Reviewer #1: Yes

4. Have the authors made all data underlying the findings in their manuscript fully available?

Reviewer #1: Yes

5. Is the manuscript presented in an intelligible fashion and written in standard English?

Reviewer #1: Yes

6. Review Comments to the Author

Reviewer #1: (No Response)

7. PLOS authors have the option to publish the peer review history of their article (what does this mean?). If published, this will include your full peer review and any attached files.

Reviewer #1: No

---

## [Editor Report · Acceptance letter]

27 Feb 2024

PONE-D-23-13722R1 

PLOS ONE

Dear Dr. Tekle, 

I'm pleased to inform you that your manuscript has been deemed suitable for publication in PLOS ONE. Congratulations! Your manuscript is now being handed over to our production team.

Kind regards, 

on behalf of

Dr. Surajit Mondal 

Academic Editor

PLOS ONE